# Chitosan–Alginate Gels for Sorption of Hazardous Materials: The Effect of Chemical Composition and Physical State

**DOI:** 10.3390/ijms25158406

**Published:** 2024-08-01

**Authors:** Erika Fekete, Emília Csiszár

**Affiliations:** 1Polymer Chemistry and Physics Research Group, Institute of Materials and Environmental Chemistry, HUN-REN Research Centre for Natural Sciences, Magyar Tudósok Körútja 2, H-1117 Budapest, Hungary; 2Department of Physical Chemistry and Materials Science, Faculty of Chemical Technology and Biotechnology, Budapest University of Technology and Economics, Műegyetem rkp. 3, H-1111 Budapest, Hungary; csiszar.emilia@vbk.bme.hu

**Keywords:** dye uptake, cationic and anionic dyes, alginate and chitosan gels, xerogels, cryogels and hydrogels

## Abstract

Chitosan, alginate, and chitosan–alginate (50:50) mixed hydrogels were prepared by freeze casting, freeze-drying, and subsequent physical cross-linking. Chitosan was cross-linked with citrate and alginate with calcium ions, while the mixed gels were cross-linked with both cross-linking agents. Both cryogels and xerogels were obtained by lyophilization and drying of the hydrogels. We investigated the effect of the chemical composition and the physical state of gels on the gel structure and sorption of model dyes. Alginate and mixed gels cross-linked with Ca^2+^ ions sorbed 80–95% of cationic dye from the solutions. The chitosan gels are primarily capable of adsorbing anionic dyes, but at near-neutral pH, their capacity is lower than that of alginate gels, showing 50–60% dye sorption. In the case of alginate gels, the dye sorption capacity of xerogels, cryogels, and hydrogels was the same, but for chitosan gels, the hydrogels adsorbed slightly less dye than the dried gels.

## 1. Introduction

In recent years, various inorganic and organic hydro-, cryo-, and aerogels have been widely used in healthcare, agriculture, and many other industries. Biogels made from natural polymers, like polysaccharides, have been gaining increasing interest [1,2,3,4,5,6] since they are biocompatible, biodegradable, inexpensive, and easy to produce. Among polysaccharides, hydro-, cryo- and aerogels prepared from chitosan and alginate (Figure 1) are being researched intensively [1,6,7,8,9,10,11,12]. Both chitosan and alginate gels and their combined forms are generally used to produce controlled drug delivery systems and scaffolds for artificial tissue culture and wound-healing coatings [1,8,10,11,13,14,15,16]. They also have several applications in water purification because they can take up significant amounts of dyes and heavy metals dissolved in wastewater [17,18,19,20,21,22,23,24,25,26,27,28,29,30,31,32,33,34,35].

Polymer hydrogels are 3D-structured cross-linked systems containing large amounts of water in their pores. The hydrogel structure prepared from polymer solution can be achieved by various methods (phase separation—induced by pH, temperature or non-solvent, cryogelation, freeze-thaw process, physical or chemical cross-linking) [2,6,9,36,37,38,39]. Various drying methods can be used to produce xero, aero-, and cryogels from polymer hydrogels [2,6,9,39]. For xerogels, the gel is dried slowly at room temperature. Generally, the resulting gel has high porosity, a large surface area, and a small pore size [39,40]. Cryogels are freeze-dried to form a super-macroporous net-structure. Despite their porous structure, they have the advantage of good osmotic and mechanical stability [40]. In the case of aerogels, the solvent is replaced by a gas without collapsing the gel structure. Supercritical extraction as a drying method usually produces an aerogel from a hydrogel without damaging the gel network. The aerogel has a particularly low density and thermal conductivity. In the case of xerogels, drying at room temperature can cause the pores to collapse due to the high capillary pressure caused by solvent removal. Freeze-drying of cryogels, on the other hand, often leads to cracks and large pores, which can be explained by the crystallization of water (or other solvents) during freezing [9].

Hydrogels can be produced from hydrophilic chitosan and alginate under relatively mild reaction conditions using inexpensive cross-linking agents [2,36]. Chitosan is prepared by deacetylation of chitin. Since deacetylation is never complete, chitosan can be considered a copolymer composed of β-(1–4)-D-glucosamine and β-(1–4)-N-acetyl-D-glucosamine units. In an acidic medium, the amine groups of chitosan are protonated, imparting a positive charge to the polymer. The alginate is a linear copolymer containing β-D-mannuronic acid (M) and α-L-guluronic acid (G) units. These elements can form a block copolymer (GGGGG or MMMMM) or an alternating copolymer (GMGMGM). The carboxylic groups in alginate are deprotonated in neutral or alkaline media, and the polymer becomes negatively charged. Chitosan is easily cross-linked with citric acid and alginate with Ca^2+^ ions [2,3,6] in aqueous media (Figure 2). Furthermore, the polyelectrolyte produced from the two polymers can also be cross-linked with these chemicals. Thus, the gels obtained by physical cross-linking do not contain harmful substances and can be used in various applications.

Chitosan and alginate are often used to prepare hydro-, cryo-, or aerogels, and micro- or nanoparticles or nanofibers containing absorbents, which can bind dyes and metal ions [17,18,19,20,21,22,23,24,25,26,27,28,29,30,31,32,33,34,35,41,42]. Adsorbents based on alginate are mainly used for cationic dyes, while chitosan-based gels bind with all types of dyes, often in combination with other polymers or fillers [41]. In many publications, the authors investigate the kinetics of sorption and give the sorption capacity of the adsorbents. These values are often compared with those shown in other publications to indicate how suitable the adsorbent produced in a particular work is. However, the data comparison is only useful if the circumstances under which the sorption test was performed (pH, adsorbent mass, dye solution concentration, adsorption time) are provided. Zhao et al., [24] for example, prepared chitosan–alginate gels with different compositions using a similar method to ours and studied the dye sorption of the gels. Their work investigated the dye uptake kinetics and pH dependence of several different gel compositions, among others. Finally, they found that their sample could bind with nearly 1500 mg/g of methylene blue, which is a good value compared to the reference values given in their table. However, to compare the efficiency of one’s sample with that of the above samples, one needs to perform lengthy browsing and calculation to do so, because it is difficult to figure out from the presented curves and tables strictly which sample the given value applies to, and what amount of adsorbent, starting solution volume, and dye concentration were used. In the absence of these, the dye removal efficiency cannot be calculated, even though it indicates what percentage of the dye content of the solution can be removed with the given adsorbent. Similar problems are also found in most of the available literature. Table 1 shows the results of some adsorption data published in the literature. They allow a comparison of the sorption efficiency of different alginate- and chitosan-based adsorbents.

In many publications, it is difficult to find under which conditions the specific adsorption capacities were measured. Hence, the data in Table 1 need to be completed and are sometimes uncertain (indicated by a question mark). We have only focused on the data for adsorbents made of alginate and/or chitosan. As most publications have investigated the effect of complex systems, we include the composition of the adsorbent tested in brackets. The data in Table 1 show that the dye removal efficiency of the pure polysaccharide adsorbents is mostly less than 90%, which, of course, may be because the contact time is often very short.

A review of the publications also revealed that the comparison of dye uptake of gels in different physical states (i.e., hydro-, cryo-, and aerogels) has yet to be investigated [43] despite this knowledge’s practical and theoretical importance. In our research, we prepared chitosan, alginate, and chitosan–alginate hybrid hydrogels without filler and other polymers using environmentally friendly methods of freeze casting and freeze-drying followed by physical cross-linking. We prepared xerogels and cryogels from the hydrogels and investigated the structure and dye-sorption capacity of the different gels.

## 2. Results and Discussion

### 2.1. Morphology of Gels

The structure of the lyophilized non-cross-linked samples is shown in Figure 3. It can be seen that a highly porous, sponge-like structure has formed after freezing and lyophilization. While interconnected pores can be observed in the pure chitosan sample (Figure 3a), the structure of the pure alginate without cross-linking (Figure 3b) is layered and lamellar. The properties, typical of both pure materials, can be observed in the 50–50% samples (Figure 3c). For hydrogels, a similar but slightly looser structure can be assumed.

The structures of the cryo- and xerogels obtained after cross-linking and drying are shown in Figure 4 and Figure 5, respectively. The SEM images reveal that the structure of the cryogels is similar to that of the lyophilized, non-cross-linked samples (Figure 4), but less regular than that of the pure chitosan and alginate gels. On the other hand, the structure of the mixed cryogels (Figure 4c,d) depends significantly on the cross-linking agent used. Similar observations can be made for xerogels (Figure 5), but their structure is even more irregular and solid.

### 2.2. Swelling of Gels

Although FT-IR spectroscopy is an excellent method for monitoring the structure of polymer systems [44,45], it is not suitable for studying the structure of hydrogels due to the large amount of bound water, so swelling tests checked the cross-linking of the samples. When cross-linking is insufficient, no gel is formed, and the samples dissolve in water instead of swelling. Figure 6 shows the different degrees of gel swelling. The swelling values of mixed gels are also presented, cross-linked with citrate and Ca ions. The results show that all four gels have significant swelling. The alginate gel swells slightly more than the chitosan gel, and the mixed gel cross-linked with Ca ions shows more significant swelling than the one cross-linked with citrate ions. This suggests that citrate ions form a more compact gel structure than Ca ions.

### 2.3. Dye Sorption of the Gels

The dye uptake of chitosan, alginate, and mixture gels was investigated in methylene blue and methyl orange solutions at different concentrations. Figure 7 illustrates the color of the residual hydrogels obtained after testing in dye solutions with a concentration of 40 mg/L. The dye uptake of the cationic methylene blue and anionic methyl orange dyes (Figure 8 and Figure 9, respectively) at 300 mg/L dye concentration is plotted as a function of time for the cryo- and xerogels.

While the alginate gel predominantly binds with the positively charged methylene blue, the chitosan gel binds with the negatively charged methyl orange. The mixed gels cross-linked with Ca^2+^ ions strongly bound to both dyes, but their methylene blue sorption (~80%) was much higher than the methyl orange sorption. Surprisingly, the methyl orange sorption capacity of the xerogels exceeded the dye uptake of chitosan. Further investigation is required to explain the results. The mixed gels cross-linked with citrate could only bind with methyl orange with an efficiency of nearly 40%.

While the alginate cryo- and xerogels can adsorb nearly 95% of the dye in the methylene blue solutions, the chitosan gels adsorb methyl orange with an efficiency of only around 60% (Figure 8 and Figure 9). The pH values of the methylene blue and methyl orange solutions were 8.34 and 7.82, respectively. These pH values confirm the measured adsorption values. In the alkaline methylene blue solution, the free carboxyl groups of alginate readily interact with the positively charged dye. However, the slightly alkaline pH of the methyl orange solution does not contribute to the significant protonation of amine groups in chitosan and the strong interaction with the negatively charged methyl orange molecules.

### 2.4. Effect of the Physical State of Gels on the Dye Sorption

It is obvious that the dried cryo- and xerogels can adsorb dyes almost equally (Figure 8 and Figure 9). We were curious to compare the dye uptake of the hydrogels with that of the dried gels, so we also performed dye uptake studies with the gels obtained immediately after cross-linking. Figure 10 shows the methylene blue adsorption of distinct types of alginate and mixed gels cross-linked with Ca^2+^ ions. Figure 11 presents the methyl orange sorption of chitosan gels.

The equilibrium methylene blue uptake was close to 95% for all three types of alginate gels, and it was slightly lower (around 80–85%) for the mixed gels. The results in Figure 10 clearly show that the dye uptake of hydrogels is practically the same as that of cryo- and xerogels. This means that the drying process can be omitted for gels prepared for application in water purification. In addition, it is essential to note that the storage of gels is easier in dried form, and the xerogels can be produced by a cheaper and simpler method than freeze-drying.

There is already some variation in the dye uptake of the different chitosan gels. The hydrogels can bind with 5–10% less methyl orange dye than the dried gels. As previously described, by lowering the pH of methyl orange solution, the dye sorption of chitosan gels is likely to be increased.

### 2.5. Comparison of the Present Gel Sorptions with Results of the Literature

In the Section 1, we pointed out that the alginate and chitosan adsorbents (gels) are often compared in terms of the specific dye uptake. The specific dye uptake values for our gels are shown in Table 2. Comparing the data with those published in the literature and given in Table 1, we find that while the specific methylene blue uptake of our alginate-containing samples is much lower than that of most alginate adsorbents (Table 1), our gels can bind with cationic dye (methylene blue), over a given time, in roughly similar proportions (expressed as %) as the reference samples. The low specific capacity values can be explained by the relatively high mass of adsorbents used in the research. Only adsorbents containing nanofibers or nanofillers show better efficiency (Table 1).

For methylene blue, the dye adsorption capacity of the 50:50 sample [24], highlighted in Table 1 and prepared similarly to our sample, is lower than that of our cryogel, which could be explained by the adsorption measurement being performed at too low a pH. While our alginate-containing gels performed similarly well in cationic dye binding as the alginate-containing adsorbents reported in the literature, the anionic dye binding of the chitosan-based gels was found to be much lower than that of the reference samples. This can be clearly explained by the fact that the pH was not adjusted adequately during our measurement. The literature results show that high-efficiency methyl orange adsorption can be achieved in highly acidic media (pH = 3).

## 3. Materials and Methods

### 3.1. Materials

KitoFlokk™ Chitosan (Mn: 100,000 g/mol, degree of deacetylation: 83%) was purchased from the Norwegian Chitosan AS (Stavanger, Norway). Sodium alginate (from brown algae, small viscosity) supplied by Sigma-Aldrich Ltd. (Budapest, Hungary). Glycerol (99.5%) and calcium chloride (99.1%) were obtained from Molar Chemicals Ltd. (Budapest, Hungary). Acetic acid (96%), methylene blue, and methyl orange, as well as tri-sodium citrate 2-hydrate (99%) were purchased from Merck Ltd. (Budapest, Hungary).

### 3.2. Sample Preparation

During the sample preparation, alginate and chitosan solutions were prepared first. To the chitosan solution, 4 g of chitosan was dissolved in 100 mL of distilled water with the addition of 1 mL of (96%) acetic acid using gentle heating (40–50 °C) and stirring. A total of 4 g sodium alginate was dissolved in 100 mL distilled water and stirred for about 1 h on a magnetic stirrer at room temperature. At the end of the solution, a yellowish viscous solution was obtained in both cases.

A total of 10 mL of the prepared solutions were pipetted into plastic sample holders. Three different compositions were used for preparing gels: chitosan (100%), alginate (100%), and chitosan–alginate (50–50%). The samples were stored at −20 °C for 24 h. Then, the lyophilization was performed on a LaboGene ScanVac Superior XS type (Scientific Laboratory Supplies, Notthingam, UK) device at −100 °C, at a pressure of 0.276 mbar, for 72 h. The samples were dry and porous (Figure 12) as their water content was sublimated.

For “green” cross-linking, 30 mL of sodium citrate (for chitosan) or calcium chloride (for alginate) solution at a concentration of 2 g/100 mL was used. The mixed gel was prepared with both cross-linking agents. In each case, the cross-linking was carried out for 20 min, and the hydrogel was washed several times with distilled water and dried in two ways. The xerogels were prepared by drying the samples for one week in a phosphorus pentoxide desiccator. The cryogels were prepared by freeze-drying in the same way as was introduced above.

### 3.3. Characterization

The structure of the dry samples was studied with a scanning electron microscope (SEM). A JEOL JSM-6380 LA-type (Jeol Ltd., Tokyo, Japan) piece of equipment was used for the tests. The non-cross-linked samples and the xero- and cryogels were examined both longitudinally and transversely.

To determine the swelling of the samples, dry samples of known weight (*W*1) were placed in distilled water for 20 min. After removal from the water, the excess water remaining on the surface of the hydrogel was gently wiped off with filter paper, and the mass of the swollen samples (*W*2) was measured. The swelling rate was calculated from the measured masses using the following formula:(1)$$Swelling\;rate\left(\%\right)=\frac{W2-W1}{W1}\times 100$$

The dye uptake of the gels was investigated in solutions with a concentration of 300 mg/L. Measurements were also performed in solutions with a concentration of 40 mg/L to demonstrate the dye uptake qualitatively. Methylene blue and methyl orange dyes were selected for characterizing gels’ cationic and anionic dye sorption, respectively (Figure 13). 

A total of 50 mL of dye solution was poured onto 0.2 g of gel (dry weight), and samples were taken from the solution at specified intervals. The absorbance of the samples was measured at 464 (MO) and 664 (MB) nm using a Unicam UV-VIS UV 500 spectrophotometer (Mettler-Toledo Ltd., Budapest, Hungary). From the absorbance values, the actual concentration of the dyesolutions was calculated using calibration equations to determine the amount of dye bound in the gel under test at a given time (Equations (2) and (3)).
(2)$$c_{t}=c_{sp}\;M_{dye}\;x\;$$
where *c_t_* is the mass concentration of the dye at time *t* (mg/L), *c_sp_* is the concentration determined from the absorbance measured with the spectrophotometer (mmol/L), *M_dye_* is the molar mass of the dye (mg/mmol = g/mol), and *x* is the dilution (−) used during sample preparation.
(3)$$q_{t}=\frac{\left(c_{i}-c_{t}\right)\;V_{i}}{m_{a}}\;$$
(4)$$q_{e}=\frac{\left(c_{i}-c_{f}\right)\;V_{i}}{m_{a}}$$
(5)$$R\left(\%\right)=\frac{\left(c_{i}-c_{f}\right)\;100}{c_{i}}$$
where *q_t_* and *q_e_* are the dye sorption capacity (mg/g) at time t and at the end of the measurement, *c_i_* and *c_f_* are the initial and final concentration of the dye solution (mg/L), *c_t_* is the concentration measured at time *t* (mg/L), *V_i_* is the initial volume of the dye solution (L), *m_a_* is the dry sample weight (g) of the adsorbent, *R* (%) is the dye removal efficiency.

The pH of the dye solutions was also determined before starting the adsorption tests. The pH of the methylene blue solution was 8.34, and that of the methyl orange solution was 7.82.

## 4. Conclusions

With the focus on sustainable development, biopolymers are also receiving increasing attention. Plastics and functional polymers made from polysaccharide-based natural polymers are already used in many areas of life. Polysaccharide gels have various healthcare, agriculture, and water purification applications. Our work has involved the preparation of chitosan, alginate, and chitosan–alginate mixed hydrogels using environmentally friendly cross-linking agents, followed by drying to produce cryo- and xerogel structures. We aimed to prepare dye-sorbing systems that could be efficiently applied to purify colored wastewater. We investigated the morphology of gels and their efficiency in dye uptake from cationic (methylene blue) and anionic (methyl orange) solutions. The results demonstrate that the alginate and the mixed gels cross-linked with Ca^2+^ ions can adsorb 80–95% of the cationic dye from the dye solution. Chitosan gels can bind with the anionic dye, but without proper pH adjustment, their capacity for dye uptake is less than that of the alginate gels; they are only capable of binding with 50–60% of the methyl orange dye. In the case of alginate gels, the dye-binding capacity of xerogels, cryogels, and hydrogels was equal. However, the hydrogels adsorbed slightly less dye for chitosan gels than the xerogels and cryogels.

## Figures and Tables

**Figure 1 ijms-25-08406-f001:**
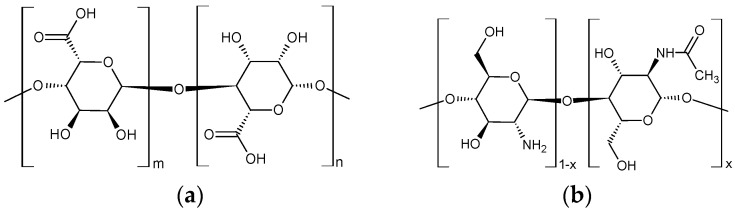
Chemical structure of alginic acid (**a**) and chitosan (**b**).

**Figure 2 ijms-25-08406-f002:**
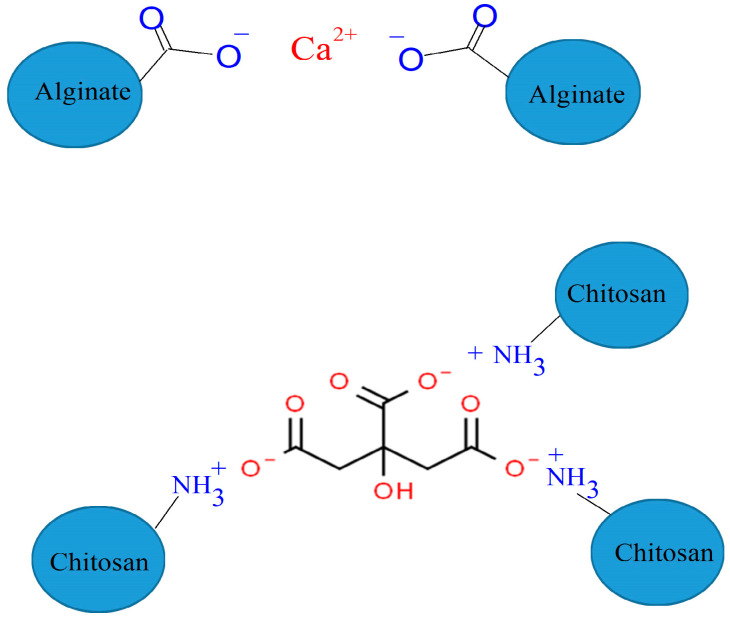
Cross-linking of alginate and chitosan.

**Figure 3 ijms-25-08406-f003:**
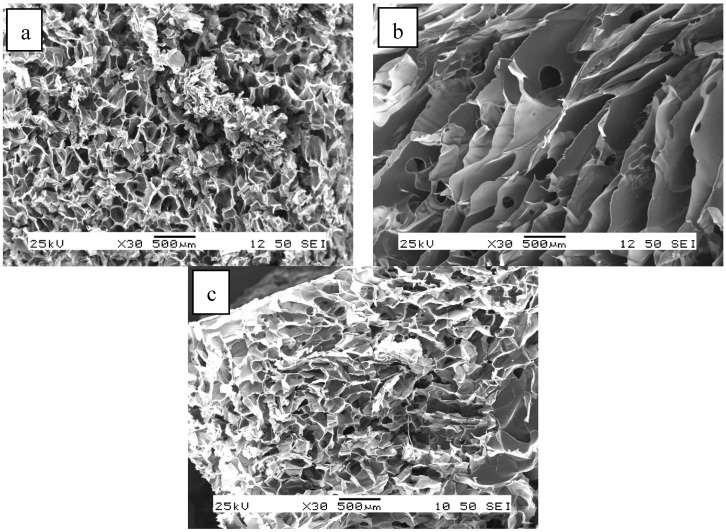
SEM images of the lyophilized non-cross-linked chitosan (**a**), alginate (**b**), and alginate–chitosan (50–50%) samples (**c**).

**Figure 4 ijms-25-08406-f004:**
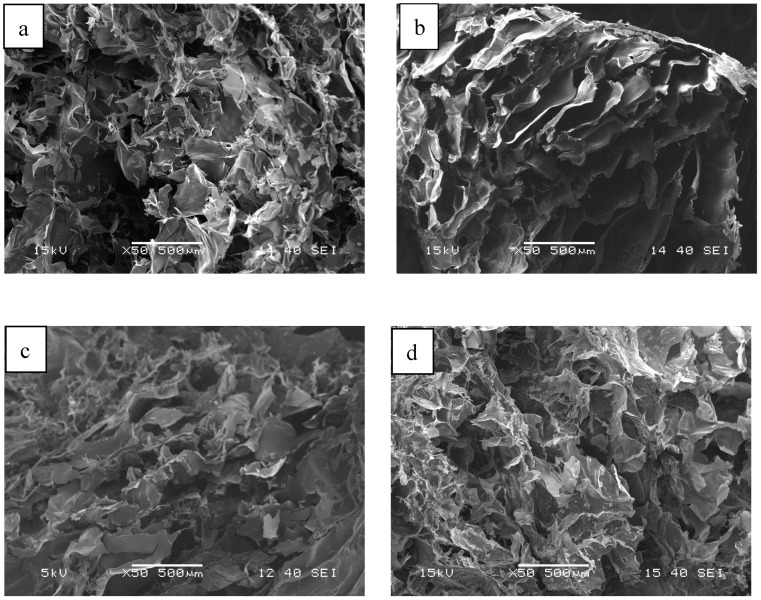
SEM images of chitosan (**a**), alginate (**b**), and alginate–chitosan (50–50%) cryogels cross-linked with citric acid (**c**) or Ca^2+^ (**d**) before drying.

**Figure 5 ijms-25-08406-f005:**
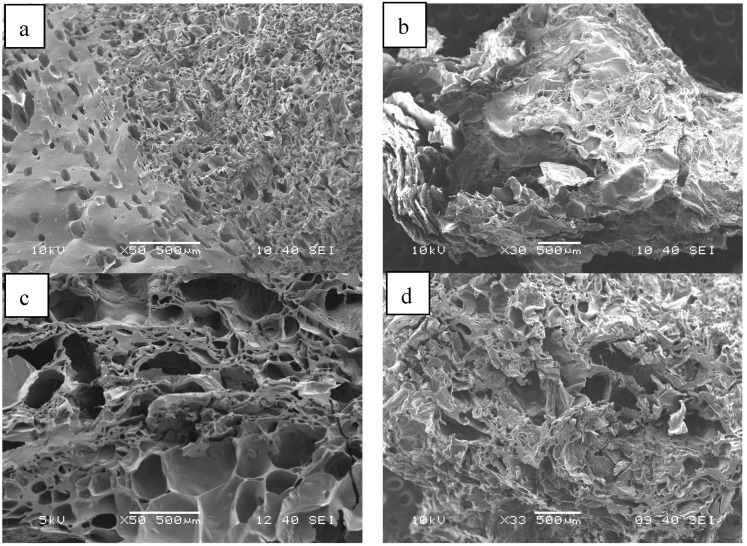
SEM images of chitosan (**a**), alginate (**b**), and alginate–chitosan (50–50%) xerogels cross-linked with citric acid (**c**) or Ca^2+^ (**d**) before drying.

**Figure 6 ijms-25-08406-f006:**
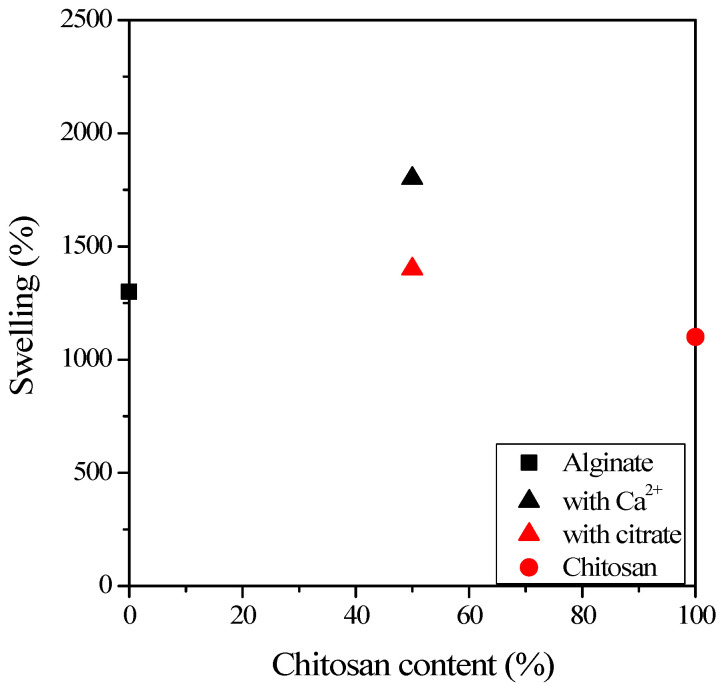
Swelling of the alginate–chitosan gels as a function of chitosan content.

**Figure 7 ijms-25-08406-f007:**
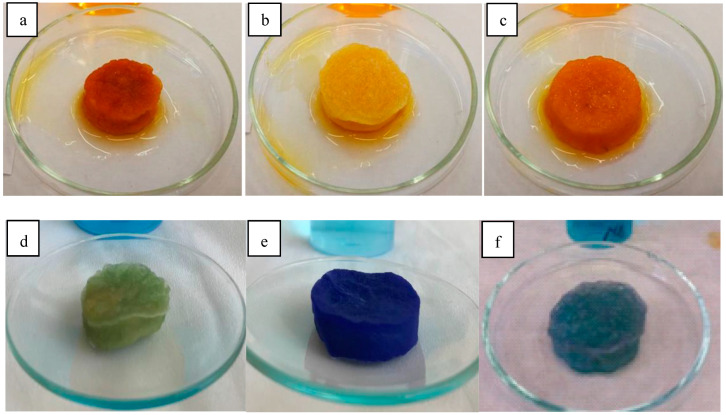
Methyl orange (upper) and methylene blue (down) sorption of the chitosan (**a**,**d**), alginate (**b**,**e**), and alginate–chitosan mixed gels (50–50%) (**c**,**f**) at 40 mg/L dye concentration for 50 h.

**Figure 8 ijms-25-08406-f008:**
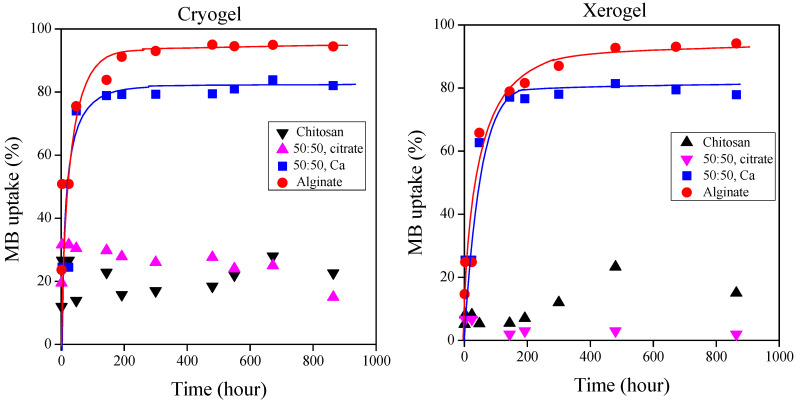
Dye adsorption of cryo- and xerogels from a 300 mg/L methylene blue solution.

**Figure 9 ijms-25-08406-f009:**
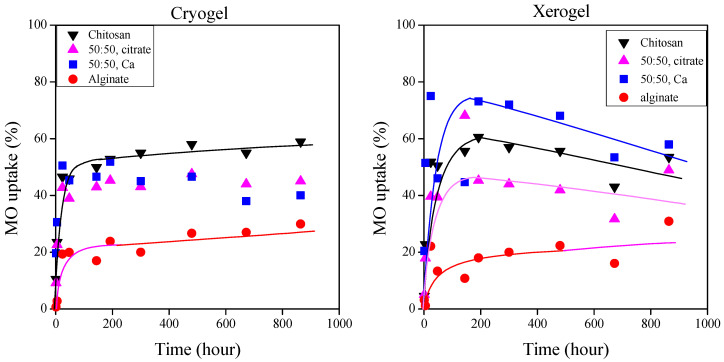
Dye adsorption of cryo- and xerogels from a 300 mg/L methyl orange solution.

**Figure 10 ijms-25-08406-f010:**
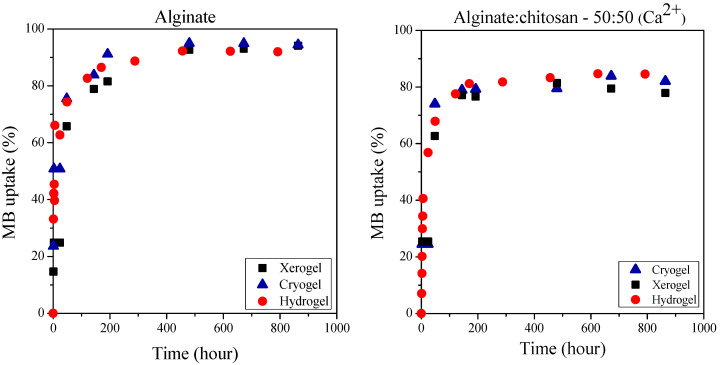
Dye adsorption of alginate and mixed xero-, cryo- and hydrogels from a 300 mg/L methylene blue solution.

**Figure 11 ijms-25-08406-f011:**
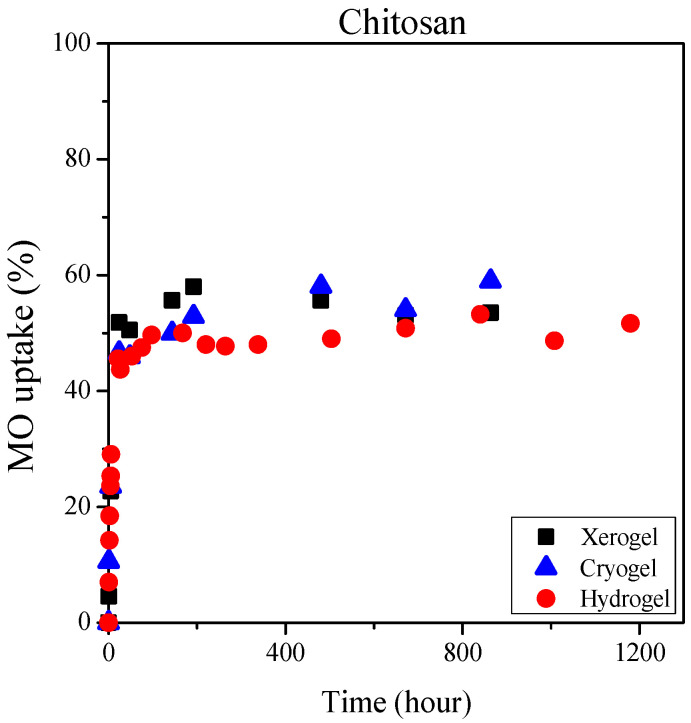
Dye adsorption of chitosan xero-, cryo- and hydrogels from a 300 mg/L methyl orange solution.

**Figure 12 ijms-25-08406-f012:**
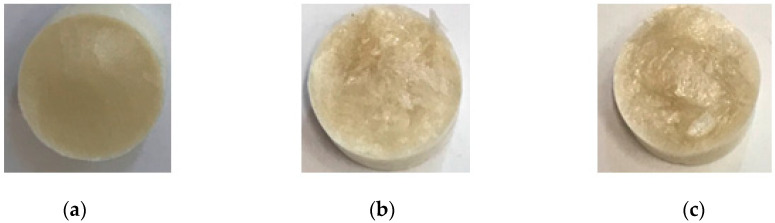
Photos taken of chitosan (**a**), chitosan–alginate (50–50%) (**b**), and alginate (**c**) samples prepared by lyophilization.

**Figure 13 ijms-25-08406-f013:**
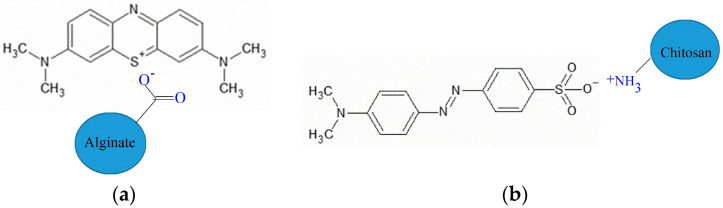
Interaction of cationic MB and anionic dyes (MO) with alginate (**a**) and protonated chitosan (**b**).

**Table 1 ijms-25-08406-t001:** Methylene blue (MB) or methyl orange (MO) uptake of different types of alginate-(SA) and/or chitosan- (CS) based adsorbents.

Composition of Adsorbent	Type of Adsorbent	Dye	Dye Removal Efficiency (%)	Dye Sorption Capacity (mg/g)	Dye Solution Concentration (mg/L)	Volume of Dye Solution (mL)	pH	Sorption Time (min)	Adsorbent Mass (mg)	Ref.
SA (SA/Polypyrrol nanotube)	microbeads	MB	~30	230	10–50	50	7	~40	50–250	[20]
SA/Graphene Oxide /Fe_3_O_4_/CS	microbeads	MB	~90	21.3	-	100	7	250–300	50	[21]
SA/Fe_3_O_4_	microbeads	MB	~77	18.3	-	100	7	250–300	50	[21]
SA/coffee grounds cellulose	microbeads	MB	~96	400.5	100	250	7	20–30	200	[22]
SA (apatite/attapulgite/alginate)	microbeads	MB	60	~139	500	100		280	25	[25]
SA/Hydroxyapatite/Graphene Oxide	microbeads	MB	~93	311	50	25	7.2	350	8	[27]
SA (Ag NPs/β-cyclodextrin/alginate)	bubbles	MB	76	-	32	-	-	30	-	[28]
SA (SA/Graphene Oxide)	quasi-cryogel beads	MB	~65	-	1–20	-	-	150	-	[29]
SA	nanofiber	MB	~95–100	2230	200–1500	50	6	50	20	[23]
**CS:SA 50:50**	**foam** **cryogel**	**MB**	**<30**	**~240**	**500**	**100**	**5.8**	**~1500**	**10**	[24]
SA covalently cross-linked by cystamine (SA-montmorillonite covalently cross-linked by cystamine)	cryogel	MB	~85	~22	25	10	-	480	10	[42]
SA	freeze-dried membrane	MB	~93		300–1000	20	8	550	10	[30]
CS	porous microbeads	MO	98.7	9.4	90	50	3	>4500	500	[35]
CS	grooved microbead	MO	98	9.14	90	50	3	4200	500	[26]
Cs	microbead	MO	~22	4.04	30	60	3		100	[31]
0.005 M CS-CaCl_2_	microbead	MO	~74	13.86	30	60	3		100	[31]

**Table 2 ijms-25-08406-t002:** Methylene blue sorption of alginate and alginate–chitosan (50–50) mixed gels and methyl orange sorption of chitosan gels measured for 800–850 h. The effect of gel physical state on the specific dye uptake capacity and dye removal efficiency.

Adsorbent/Dye	pH	Cryogel		Xerogel		Hydrogel	
		R (%)	q_e_ (mg/g)	R (%)	q_e_ (mg/g)	R (%)	q_e_ (mg/g)
Alginate/MB	8.3	95	71.3	94	70.5	92	68.5
50:50 (Ca^2+^)/MB	8.3	83	62.2	79	59.2	85	63.7
Chitosan/MO	7.8	56	42.0	54	40.5	52	38.5

## Data Availability

The original data presented in the study are openly available in [REAL repository] at [https://real.mtak.hu].

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
