# Peer review of "Chitosan–Alginate Gels for Sorption of Hazardous Materials: The Effect of Chemical Composition and Physical State"

_ijms, 2024, doi:10.3390/ijms25158406_

Round 1
Reviewer 1 Report
Comments and Suggestions for Authors
In this article, the effects of the preparation, chemical composition and physical state of chitosan, alginate and chitosan - alginate (50:50) mixed hydrogels on gel structure and model dye adsorption were studied. They did a good job. But there are some problems associated with this work.
1. When discussing the problem with the various testing methods, please provide specific examples or statistics to highlight the severity of the problem. This will give more weight to your argument.
2. Please add more experimental characterization data to enrich your article, such as mechanical strength of gel, IR spectrum of gel, adsorption-desorption curve of gel, pore size distribution, etc.
3. Please add more experimental test data to enrich your article, such as the adsorption effect of gel on dyes of different concentrations, adsorption thermodynamics, adsorption effect at different pH values, cyclic adsorption, etc
4. In order to avoid confusion among readers, please divide the experimental methods in "Results and Discussion" into several parts and put them in the experimental section. Provide a detailed description of the experimental method.
5. The explanation of the adsorption mechanism of cationic dyes by gels in this article is not clear enough, so it is suggested to give a clearer explanation of the adsorption of cationic dyes by gels combined with experimental data.
6. Please proofread thoroughly and remove or revise sentences with repetitive meanings to perfect the essay and correct any grammar, formatting, typos or clumsy diction. All graphics, schemes, and tables should be inserted into the text. Please check each title carefully and fill it in according to the journal requirements.
7. The time selected for the adsorption kinetics experiment of gel on dye in this article is relatively long, so it is suggested to shorten the time to 0-1000min
8. There are some recent publications about porous organic polymers for heterogeneous catalysis, the author may cite them to enrich the introduction and/or compare the catalysis properties, for example: J Centr. South Univ. 2020, 22:1247; Polymer, 2021, 233: 123194; Polymers, 2022, 14, 1900.
Comments on the Quality of English Language
In this article, the effects of the preparation, chemical composition and physical state of chitosan, alginate and chitosan - alginate (50:50) mixed hydrogels on gel structure and model dye adsorption were studied. They did a good job. But there are some problems associated with this work.
1. When discussing the problem with the various testing methods, please provide specific examples or statistics to highlight the severity of the problem. This will give more weight to your argument.
2. Please add more experimental characterization data to enrich your article, such as mechanical strength of gel, IR spectrum of gel, adsorption-desorption curve of gel, pore size distribution, etc.
3. Please add more experimental test data to enrich your article, such as the adsorption effect of gel on dyes of different concentrations, adsorption thermodynamics, adsorption effect at different pH values, cyclic adsorption, etc
4. In order to avoid confusion among readers, please divide the experimental methods in "Results and Discussion" into several parts and put them in the experimental section. Provide a detailed description of the experimental method.
5. The explanation of the adsorption mechanism of cationic dyes by gels in this article is not clear enough, so it is suggested to give a clearer explanation of the adsorption of cationic dyes by gels combined with experimental data.
6. Please proofread thoroughly and remove or revise sentences with repetitive meanings to perfect the essay and correct any grammar, formatting, typos or clumsy diction. All graphics, schemes, and tables should be inserted into the text. Please check each title carefully and fill it in according to the journal requirements.
7. The time selected for the adsorption kinetics experiment of gel on dye in this article is relatively long, so it is suggested to shorten the time to 0-1000min
8. There are some recent publications about porous organic polymers for heterogeneous catalysis, the author may cite them to enrich the introduction and/or compare the catalysis properties, for example: J Centr. South Univ. 2020, 22:1247; Polymer, 2021, 233: 123194; Polymers, 2022, 14, 1900.
Reviewer 2 Report
Comments and Suggestions for Authors
This study is interesting and covers the important topic of hazardous waste treatments. The following are some comments and suggestions for improvement:
1. In this study, chitosan was crosslinked with citrate and alginate with calcium ions; however, the specific mechanisms and conditions for these processes are not detailed. How were the crosslinking densities and efficiencies measured or controlled?
2. Chemical composition and physical properties (such as porosity) of a gel may have significant effects on dye adsorption. The authors should at least perform Fourier transform infrared spectroscopy (FTIR) analysis to identify the chemical structural differences between gels to clarify this issue.
3. Information on the physical and mechanical properties of the gel, such as compressive strength and porosity, is needed to determine, not only swelling ratio.
4. While the study provides some data on the sorption capacities of the gels, it lacks a discussion on the possible mechanisms behind the sorption of dyes. The authors need to analyze which type of adsorption kinetics and Langmuir isotherm models fit the experimental data.
5. The adsorption data should be compared with previous similar studies, and tables may be needed to better illustrate why the chitosan-alginate (CS-AL) gel developed in this study is superior in dye adsorption to other CS-AL gels.
The authors need to cite some literature to expand the application potential of this study, such as Ca2+-crosslinked AL/CCS gels for adsorption of natural dye anthocyanins (10.1016/j.carbpol.2022.120133; 10.1016/j.heliyon.2023.e18879).
Comments on the Quality of English LanguageMinor editing of English language required
Round 2
Reviewer 2 Report
Comments and Suggestions for Authors
Accept
Comments on the Quality of English LanguageAccept